# Influence of Casting Variables on Release Kinetics of Orally Disintegrating Film

**DOI:** 10.3390/foods13091418

**Published:** 2024-05-05

**Authors:** Jang-Ho Shin, Jung-Ah Han

**Affiliations:** 1Department of Food Service Management and Nutrition, Sangmyung University, Seoul 03016, Republic of Korea; sjh1219@naver.com; 2Department of Food and Nutrition, Sangmyung University, Seoul 03016, Republic of Korea

**Keywords:** orally disintegrating film, casting thickness, vitamin C, dissolution pattern, release mechanism

## Abstract

As a new form for supplying vitamin C, orally disintegrating films (ODFs) were developed C based on hyaluronic acid (HA) under varying casting conditions and the properties were analyzed. The films with different thicknesses (2, 3, and 8 mm, for CT2, CT4, and CT8, respectively) were produced by adjustments made to casting height. Two types of 8 mm thick ODFs produced by single or double casting (4 + 4 mm for CTD4+4) methods were also compared. As film thickness increased, water vapor permeability and tensile strength also increased. Even at equal thickness, manufacturing with double casting exhibited a stronger texture and reduced disintegration compared to single casting. All ODFs met the World Health Organization’s recommended daily vitamin C intake (45 mg/day) with a single sheet. Films showed over 80% dissolution in various solvents, adhering to the Hixson–Crowell cube root law, indicating vitamin C release occurred via porous penetration of the eluate. For CT2, CT4, and CTD4+4, vitamin C release was primarily governed by diffusion within the gel matrix and HA erosion. However, for CT8, HA erosion-induced release somewhat dominated. Based on the sensory test, it seems desirable to adjust the thickness of the film to 2 or 4 mm, because a thickness greater than that increased the foreign body sensation due to prolonged residence in the oral cavity.

## 1. Introduction

Vitamin C is a water-soluble vitamin that plays important roles in maintaining cell metabolism, division, and proliferation, and is a major antioxidant in organic organisms [1]. However, the human body cannot synthesize vitamin C or store excess vitamin C in the body, and in order to obtain additional benefits such as antioxidant effects in addition to vitamin C’s basic role as a nutrient, it is needed to additionally take vitamin C supplements in various formulations [2]. Drugs or functional foods, including vitamin C, are supplied in the form of liquids, capsules, powders, tablets, etc. However, these formulations must be swallowed with water or chewed with teeth. Therefore, it can be difficult to use for the elderly and those who suffer from other physical or mental issues that affect swallowing [3]. An orally disintegrating film (ODF) is a dosage form in the form of a small and thin film that is rapidly degraded on the tongue or in the oral cavity, which has recently attracted attention because it has the advantage that it can be conveniently taken orally without water. 

As base materials for ODFs, several water-soluble polymers have been reported such as hydroxypropyl methylcellulose [4], hydroxypropyl cellulose [5], starch [4,6], and gelatin [6]. Recently, hyaluronic acid (HA) has been in the spotlight as a natural polymer for ODF because it also has excellent hydrophilicity and gel-forming ability, and HA allows transparent and flexible ODFs to be manufactured without any plasticizer [7]. For example, HA-based ODFs have been reported to treat oral ulcers [8] or relieve dry mouth symptoms [9]. 

The important properties of film, such as mechanical properties and water-holding capacity, can be affected by the film thickness [10]. Specifically, the thickness of the ODF should strike a balance between being thin enough to adhere comfortably to the oral mucosa and being thick enough to facilitate handling [3]. Incorporating functional components in the ODF may lead to an increased amount of active substances per unit area with increasing film thickness. However, this may also result in a heightened perception of a foreign body sensation in the mouth. Therefore, it is crucial to carefully adjust the thickness of the ODF to attain the desired properties. 

The oral sublingual mucosa attached to ODF has the advantage of excellent permeability due to its relatively thin membrane and large veins, providing rapid absorption and immediate bioavailability of active ingredients [11]. Thus understanding the characteris-tics of ODF dissolution in the body can be considered as necessity for its utilization. Okamoto et al. [12] compared penetration rate of lidocaine through excised oral mucosa from hamster cheek pouch and the in vitro release rate of LC from film dosage forms with hydroxypropylcellulose as a film base, and they concluded that the in vitro dissolution study is a useful tool to predict the penetration rate taking the unionized drug fraction into consideration. A few studies about differences in dissolution profiles by several layed film were reported. For examples, Preis et al. [13] manufactured bilayered ODF composed of drug loaded layer and backing layers without drug by various polymers and evaluated the dissolution test. Abu-Huwaij et al. [14] prepared bi- and triple layer patches with different penetration enhancer as a buccal mucoadhesive delivery system for systemic delivery of lidocaine and reported that the release mechanism was different by the kind of penetration enhancer. However, there are no reported studies on the release characteristics of films manufactured by varying the casting thickness and frequency using the same film forming material. In this study, an ODF containing vitamin C was manufactured using HA under various casting conditions (different thicknesses and single vs. double casting). Also, the dissolution profile and physicochemical properties were analyzed and compared with the aim of providing fundamental data to broaden the application of this product in the food and pharmaceutical industries.

## 2. Materials and Methods

### 2.1. Materials

Hyaluronic acid (Mw 8 × 105 Da, Jinwoo bio, Seoul, Korea), vitamin C powder (DSM UK Ltd., Derbyshire, UK), sucralose (Shandong Kanbo biochemical technology Co., Ltd., Dongying, Shandong, China), and anhydrous ethanol (Duksan pure chemicals Co., Ltd., Ansan, Gyeonggido, Korea) were used as raw materials for film production. Ascorbic acid (Sigma Co., St. Louis, MO, USA), acetonitrile (Duksan Pure Chemicals Co., Ltd.), metaphosphoric acid (Kanto Chemical Co., Inc., Tokyo, Japan), and DPPH radicals (Sigma Co.) were purchased to measure vitamin C content and antioxidant activity. NaCl, KH2PO4 and Na2HPO4 (Kanto Chemical Co., Inc.), HCl and NaOH (Samchun Pure Chemical Co., Ltd., Pyeongtaek, Gyeonggido, Korea), sodium acetate (Daejung Chemicals & Metals Co., LTD., Siheung, Gyeonggido, Korea), and acetic acid glacial (JT Baker, Phillipsburg, NJ, USA) were also prepared. 

### 2.2. Preparation of Vitamin C-Loaded Hyaluronic Acid-Based ODFs

The contents of HA, vitamin C, and sucralose for the film-forming solution were set through preliminary experiments and then prepared as follows. First, 3 g of HA was added to 10% ethanol (100 mL) and stirred at room temperature for 24 h. Vitamin C (4 g) and sucralose (13.7 mg) were added to distilled water (12 mL) and stirred at room temperature for 30 min. After mixing the prepared two solutions and stirring at room temperature for another 30 min, they were then sealed and stored for 24 h at 4 °C to remove bubbles. The prepared film solution was cast in different thicknesses using a casting machine (COAD.411, Ocean Science, Uiwang, Gyeonggido, Korea), and then dried in a thermo-hygrostat (TH-PE-065, Jeio Tech Co., Daejeon, Korea). 

The casting thickness was set to 4 types of 2, 4, 4 + 4, and 8 mm through many preliminary experiments. To produce films of the same size (15 × 20 cm), 50 mL film solution was cast to 2 mm (CT2), 150 mL solution was cast to 4 mm (CT4), and 300 mL film solution was cast to 8 mm (CT8). The 4 + 4 mm film (CTD4+4) was prepared by double casting, viz., first 150 mL of film solution was cast at a thickness of 4 mm, dried for 4 h, and then an additional 150 mL of film solution was cast on the dried film surface by stacking, and dried for 20 h. All films were stored for 24 h in a precision thermo-hygrometer (DTM-321, Tecpel Co., Ltd., Taipei, Taiwan) maintained at 25 ± 1 °C and 44 ± 1 RH%. Then, the size of one film was cut by 6 cm^2^ (2 cm × 3 cm). In the case of conducting an experiment requiring a specific size, it was separately cut to a size suitable for the experimental method.

### 2.3. Appearance and Microstructure

Referring to the method of Garcia et al. [6], the appearance of the film was evaluated based on three items: uniformity (uniformity of color and thickness, formation of insoluble particles), film forming ability (maintaining continuity of the film after casting and drying), and ease (ease of separation from supporter after drying). For the microstructure, the film was cut into dimensions of 2 × 5 mm, fixed with carbon tape, and gold ion-coated for 180 s. The surface and cross-sectional area of films were examined using a scanning electron microscope (JSM-560 OLV, Tokyo, Japan) at 30 kV. Sections were photographed at ×500 and ×1000, respectively.

### 2.4. Thickness, Weight, Surface pH, and Transparency

The thickness of each film was measured more than 5 times on various parts using a dial caliper (D15HA, Mitutoyo Co., Kawasaki, Japan). The weight was measured using a precision electronic scale (E12140, Ohaus Co., Pine Brook, NJ, USA).

The surface pH of the film was measured by referring to the method of Garcia et al. [6]; in total 0.5 mL of phosphate buffer solution (pH 6.8) was added to the surface of the film and measured after 30 s using a pH meter (Testo 206, Testo SE & Co., KGaA, Lenzkirch, Germany).

The transparency of films was measured using a spectrophotometer (UV-160A, Shimadzu Co., Kyoto, Japan) by measuring the % transmittance at 600 nm, according to the method of Han and Floros [15]. 

The transparency per unit light path length was calculated from Equation:Transparency=(X600)/y
where X600 is the % transmittance at 600 nm and y is the film thickness.

### 2.5. Mechanical Properties

To measure the mechanical properties, the film was cut into 2 × 7 cm size, and the initial cross-sectional area (A_0_, mm^2^) was obtained by measuring the thickness of the film using a dial caliper (D15HA, Mitutoyo Co., Kawasaki, Japan). The tensile strength (TS) and elongation at break (EB) of the film were measured using a texture analyzer (TA-XT2i, Stable Micro Systems, Surrey, UK) according to ASTM D882-10 [16]. Under the condition that the distance between the initial grips (L_0_) is 50 mm and the movement speed of the grips is 50 mm/min, the maximum tensile load (W_max_, N) and the moving distance (L, mm) of the grips are obtained, and substituted into the equation.
$$Tensile\;strength\;(MPa)=\frac{W_{max}}{A_{0}}$$
$$Elongation\;break\;(\%)=\frac{L}{L_{0}}\times 100$$

### 2.6. Moisture Content and Water Vapor Permeability

The moisture content of the film was calculated as the percentage of weight difference before and after drying in a drying oven (C-DO02, Changshin Scientific Co., Hwaseong, Korea) for 48 h. The water vapor permeability (WVP) of the film was measured according to the method described by Cao and Song [17]. A film (3 × 3 cm) was fixed to the inlet (2 × 2 cm) of a moisture-permeable cup filled with distilled water (18 mL), sealed, and then placed in a thermo-hygrostat set at 25 °C and 50 RH% (TH-PE-065, Jeio Tech Co., Daejeon, Korea), and the weight of the cups was measured at 1 h intervals for 8 h. The WVP was calculated by following the equations:WVPR=Slope/(Film area)
WVP=(WVPR×L)/Δp
Δp=610.7×10^(7.5T/(237.3+T))×(1−RH/100%)

Here, the slope is the weight reduction rate of the moisture-permeable cup over time, the film area is the area of the film where moisture moves (mm^2^), L is the average thickness of the film (mm), and Δp is the difference in water vapor between the inside and outside of the cup. T is the temperature (°C) and RH is the relative humidity (RH%).

### 2.7. In Vitro Disintegration Time and In Vivo Disintegration Rate

The in vitro disintegration time of each ODF was measured according to the method of Steiner et al. [18]. That is, after fixing a film (3 × 4 cm in size) to a frame with an empty center (2 × 3 cm in size), a weight (10 g) and distilled water (0.9 mL) were simultaneously applied to the center of the film. The time for the weight to penetrate the film and fall to the floor was measured. The in vivo disintegration rate of the ODF was evaluated by 25 panelists in their 20 s–60 s. The prepared film with a three-digit number using a random number table was presented to the panel. The film was asked to be attached on the roof of the mouth and the melting speed was subjectively evaluated using a 7-point scale method: 1 point was ‘very slow’, 4 points was ‘neither slow nor fast’, and 7 points was ‘very fast’.

### 2.8. Measurement of Vitamin C Amounts in a Sheet of Film

The amount of vitamin C in each film was measured according to the method of Yan et al. [19]. The analysis was performed using an HPLC system (Ultimate 3000 HPLC system, Thermo Fisher Scientific Inc., Waltham, MA, USA) equipped with an Acclaim™ 120 C18 column (5 μm, 120 Å; 4.6 mm × 150 mm, Thermo Fisher Scientific Inc., Waltham, MA, USA). For measurement, each film was dissolved with 50 mM potassium dihydrogen phosphate and then filtered with a 0.45 mm syringe filter (Millipore, Burlington, MA, USA). The mobile phase was KH_2_PO_4_:acetonitrile (95:5, *v*/*v*) and detection was achieved at 254 nm. For the measurement conditions, a 0.8 mL/min flow rate, a column temperature of 25 °C, and a 10 μL injection volume were applied. Standard solutions were prepared and used immediately before analysis at concentrations of 0.25, 0.5, and 1.0 mg/mL using solutions of ascorbic acid and 5% metaphosphoric acid.

### 2.9. In Vitro Dissolution Behavior

In order to find out the vitamin C elution pattern in the ODF in each digestive fluid of the human body, vitamin C’s elution behavior was measured by the method of Huang et al. [20]. The eluate used in the dissolution test was simulated gastric fluid (SGF, pH 1.2), simulated intestinal fluid (SIF, pH 6.8), acetate buffer (AB, pH 4.5) as an artificial intestinal fluid, and simulated saliva (SSF, pH 6.8) prepared by the method of Peh and Wong [21] listed in the US Pharmacopoeia, and distilled water. The composition of each elution is shown in Table 1. 

Each film was attached to the bottom of the beaker, and then each elution solution (50 mL) was added and incubated in a shaking incubator (SI-600R, JEIO TECH Co., Ltd., Daejeon, Korea) at 37 ± 1 °C at 100 rpm for 60 min. Depending on the time, 0.5 mL of the eluate was taken, mixed with the same amount of 10% metaphosphoric acid solution, and then injected into the HPLC (Dionex Ultima 3000 HPLC system, Thermo Fisher Scientific Inc., Waltham, MA, USA) system. The elution amount of vitamin C in the film was measured. After taking the sample, the same type of eluate was added to the beaker to maintain the volume of the eluate at 50 mL during the experiment.

### 2.10. Model Formula Application for Dissolution Mechanism Analysis

In order to understand the dissolution mechanism of vitamin C contained in each film, the dissolution pattern was analyzed by applying the model formula below.
$$Zero-order\;kinetics:M_{0}-M=K_{0}\times t$$
$$Hixson–Crowell\;cube\;root\;law:M_{0}^{⅓}-M^{⅓}=K_{⅓}\times t$$
$$Higuchi\;model:M_{0}-M=K_{1/2}\times t_{1/2}$$

Here, M_0_ is the content of vitamin C (mg) per sheet of ODF (2 × 3 cm), M is the content of unreleased vitamin C after a certain period of time (mg), and K_0_, K_1/3_, and K_1/2_ are the elution rate constant of each model, and t is the elution time. In addition, the diffusion pattern of vitamin C in the ODF was analyzed by applying the Korsmeyer–Peppas model below [22].
$$F=M_{t}/M=K_{m}\times t^{n}$$

In the above formula, F is the elution rate of vitamin C, Mt is the amount of vitamin C eluted at time t (mg), M is the total amount of vitamin C in the film (mg), Km is the dissolution rate constant, and n is the dissolution index. 

### 2.11. Evaluation of Sensory Properties

The sensory test of the film was evaluated by conducting a preference test on 30 panelists in their 20 s–60 s. Each film (2 × 3 cm), with a three-digit number using a random number table, was provided with questionnaires and instructions, and the six sensory properties (appearance, sourness, sweetness, comfort, mouth feel properties, and overall preference) of the films were asked to be evaluated using a 7-point scale. The sensory test was performed with the approval of the bioethics committee of the affiliated institution (IRB-SMU-S-2020-1-002). 

### 2.12. Statistical Analysis

All data were determined at least three times. The measurement results were subjected to analysis of variance (ANOVA) using SPSS (Statistical Package for Social Sciences, Version 23.0, IBM-SPSS Inc., Chicago, IL, USA). The significant difference between each film was tested at *p* < 0.05 with the Duncan test.

## 3. Results and Discussion

### 3.1. Appearance and Microstructure 

After casting the film solution, no undissolved solids or crystals were found, and a uniform film was formed. After drying, when the film was removed from the supporter (OHP film), except CT2 which showed some cracking or tearing, other ODFs could be easily removed from the supporter. Usually, the absence of pores and surface uniformity is a condition of the good quality of films [23]. As shown in Figure 1a, CT2 had a smooth face, whereas some bubbles were observed on the surface of CT4, and irregular pits were observed on the surface of CTD4+4. The irregular pits and air bubbles were also observed inside of CTD4+4 and CT8. Small air bubbles formed on the surface and inside of the film are thought to be due to some air being introduced into the film solution during the casting process. In the case of CT2, due to its thin thickness, air was naturally degassed during the drying process; however, for the ODFs with thicker thickness, it seems that the surface was dried first, and the air bubbles inside were not degassed, resulting in the bubbles remaining inside of the dried film. 

The scanning electron microscope image of the ODF is shown in Figure 1b. The cross-sectional microstructure of the film is shown in Figure 1. As shown in Figure 1, CT2 and CT4 showed smooth cross-sections. In the case of CTD4+4, interlayer straight connecting lines (red arrows in Figure 1) were observed, and the cross-section was not uniform. This is similar to the findings of Preis et al. [13] and indicates that the second film solution cannot spread and distribute uniformly within the first layer. While CT8 showed a uniform cross-sectional microstructure compared to CTD4, some cracks were observed inside the film.

### 3.2. Thickness, Weight, Surface pH, and Transparency

The thickness and weight of the ODFs are shown in Table 2. The thickness of an ODF is one of the important factors in evaluating it, as it is directly related to the content and uniformity of the active substance contained in the film [24]. As the casting thickness increased to 2, 4, 4 + 4, and 8 mm, the film thickness and weight also significantly increased to 0.09, 0.14, 0.19, and 0.24 mm and 115.40, 124.73, 169.03, and 205.00 mg, respectively. Comparing CTD4+4 and CT8, which had the same casting thickness with a different number of castings, there was also a significant difference; CT8 (0.24 mm, 205.00 mg) showed significantly higher values than CTD4+4 (0.19 mm, 169.03 mg). Despite the complete drying of the first film layer prior to casting the second layer, there is a possibility that the solvent from the second layer caused a slight dissolution of the first layer before the complete casting of the second layer, leading to an uneven bilayered film, as proposed by Preis et al. [21]. This phenomenon could also account for the reduced thickness of CTD4+4 compared to CTD8.

Thickness, which is influenced by the water-holding capacity of the film material, is known to affect the physical properties of films such as tensile strength, elongation, and water vapor permeability [10]. Particularly, in starch-based edible films, glycerol is commonly used as a plasticizer, and its water-holding capacity has been reported to increase the water absorption of the film, leading to swelling and an increase in thickness, especially at high glycerol concentrations [25]. While an increasing film thickness can result in containing a higher amount of active substance per unit area, it is important to adjust the thickness of the film to achieve the desired properties since the film characteristics can be affected by its thickness. 

Table 2 displays the surface pH values of the ODF. As the casting thickness increased, the surface pH values significantly decreased. Specifically, the highest value of 4.05 was observed for CT2, while the lowest values ranging from 3.51 to 3.56 were observed for CTD4+4 and CT8. The pH of the films measured in this study was similar to the pH range (3.6–3.9) of vitamin C-loaded pectin films reported by Pérez et al. [26]. Jyoti et al. [27] reported that sour-tasting ingredients such as citric acid, ascorbic acid, and tartaric acid can stimulate salivation by activating the salivary glands as gustatory stimuli, promoting the dissolution of oral strips in the oral cavity. Humphrey and Williamson [28] reported that sour and sweet tastes can stimulate saliva secretion in patients with dry mouth syndrome. Kweon and Han [9] conducted a salivation stimulation experiment on the elderly and reported that the oral dissolving film made of hyaluronic acid actually promoted salivation in the oral cavity due to the high water retention capacity of hyaluronic acid itself. Therefore, for the HA-based ODF containing vitamin C developed in this study, not only the effect of supplying vitamin C, but also the effect of relieving dry mouth symptoms can be expected. 

In the case of transparency, as the film thickness increased, it significantly decreased; CT2 showed the highest value of 20.05, and CT8 showed the lowest value of 8.04. This is a result similar to the yellowness, and as the thickness increases, the color of the film overlaps and the transparency seems to decrease. In general, transparency is a necessary factor for the visibility of food in decorative films for packaging, but in the case of some light-sensitive foods or raw materials, the opacity of the film acts as a barrier to light, so it seems necessary to properly adjust the transparency and opacity of the film [17].

### 3.3. Mechanical Properties

Table 3 shows the mechanical properties of ODFs manufactured with different thicknesses. In the single-casted film, as the casting thickness increased, the TS also increased, with CT2 showing the lowest value at 6.28 MPa and CT8 showing the highest value at 7.99 MPa. Kweon and Han [9] reported that the TS of the HA film was proportional to the concentration of HA in the film. In this study, because the higher casting thickness means an increase in the HA content in a film, TS was proportional to the casting thickness. According to the number of castings, CTD4+4 showed a higher value (16.61 MPa) than CT8. 

The elongation at break (EB) was found to increase as the thickness increased in the range of 25.34–54.08%, but there was no significant difference between CT4 and CT8. According to the number of castings, the double casting (CTD4+4) showed higher TS and EB values than the single casting (CT8). It is generally known that the TS and EB of a film are inversely proportional because an increase in the EB of a film means that the film stretches well and the strength when the film breaks is lowered [29]. However, CTD4+4 showed the highest TS and EB values, and it could imply that the produced delicate interstitial spaces inside film (Figure 1b) conferred distinctive properties upon CTD4+4.

To compensate the disadvantage of easy breakage in starch or protein-based materials commonly used for ODF, it is necessary to incorporate suitable plasticizers that could improve the film’s properties by enhancing the flexibility of polymer chains [17]. However, when manufacturing films using HA as a raw material, it appears possible to adjust properties such as TS and EB without the addition of such plasticizers by controlling the thickness and casting number.

### 3.4. Moisture Content (MC) and Water Vapor Permeability (WVP)

The MC and WVP of ODFs are presented in Table 2. As for the MC, CT2 showed the lowest value at 16.62% and CT8 showed the highest value at 20.59%, showing a tendency to increase with thickness. Due to the strong water-holding capacity of HA, moisture can be retained within the HA-based film [30], so as the film thickness increases, a greater amount of HA can hold moisture, resulting in an increase in the film’s MC. For the two films manufactured with the same thickness but different casting times, the MC of CTD4+4 was lower than that of CT8, because during the second casting and drying of CTD4+4, a certain amount of moisture could be evaporated. Borges et al. [31] found that the MC is a factor that has a great influence on the quality characteristics of the film. If the MC is too low, the film could be easily broken, and the space between the polymer chain bonds constituting the film is reduced, resulting in an increasing water permeability and a delayed disintegration of the film. On the contrary, when the MC was too high, stickiness was generated on the film surface. In this study, CT2 showed some cracking, but CT4 and CTD4 were confirmed to have flexibility as well as non-stickiness, so the adequate moisture content of the HA-based ODF is considered to be about 18–20%. 

Water vapor permeability (WVP) is the property of a film to attract or evaporate moisture from the surrounding environment [32]. WVP is an important factor of ODF that determines the prevention or reduction in moisture transfer from the surroundings during storage [33], and various factors such as film thickness and humidity could affect the WVP [32]. The larger the WVP is, the greater the spread rate of water molecules in the film, which can lead to greater decay or deformation [34].

As for the WVP of the ODF developed in this study, CTD4+4 was the highest with 11.73 g mm/m^2^ h kPa, followed by CT8, CT4, and CT2 with 11.36, 7.07, and 5.67 g mm/m^2^ h kPa, respectively, showing a tendency to increase as the casting thickness increases (Table 2). This is consistent with the previous research results that the WVP increases as the thickness of a film based on a hydrophilic material increases [35]. The WVP of films made of hydrophilic properties such as chitosan [36] and cellulose [37] increased with the moisture content of the film. 

As for the 8 mm ODF, CTD4+4 had a lower thickness and moisture content and a higher WVP than CT8. As shown in Figure 1, it is presumed that the CTD4+4 mm film has a movement of moisture through the space existing between the two layers due to the laminated structure [38]. Sharma and Singh [39] reported that the WVP should be reduced as much as possible to improve the quality of edible films. In this study, the WVPs of CT2 and CT4 were 5.67–7.07 g mm/m^2^ h kPa, which was similar to the value of apple film reported by Sablani et al. [40] (4.20–7.56 g mm/m^2^ h kPa), and higher than the potato peel film (2.99–5.30 g mm/m^2^ h kPa) reported by Kang and Min [41], but lower than that of casein-based edible films (7.91 g mm/m^2^ h kPa) reported by Chick and Ustunol [42]. The relatively high WVPs for CTD4+4 and CT8 mean a high water absorption rate, and could affect the stability of vitamin C by increasing the moisture content and dissolved oxygen content of the film [43]. Therefore, it could be desirable to lower the WVP by adjusting the casting thickness to 4 mm or less in the produced ODF for the purpose of supplying vitamin C.

### 3.5. In Vitro Disintegration Time and In Vivo Disintegration Rate

As shown in Figure 2, the in vitro DT of CT2 and CT4 was 1.53–1.70 min, showing no significant difference, but it increased significantly as the film thickness increased, and CTD4+4 and CT8 were 2.27 and 7.10 min, respectively. DTs of films could be varied and controlled by film materials, and the DTs clearly correlated with the thickness of films because the immersion of water into the film was likely to depend on the film thickness [44]. The in vivo disintegration rate (DR) was the fastest at 5.12 for the CT2 film and the slowest at 2.92 for the CT8 film, indicating that the DR slowed down as the thickness increased. 

The DT of oral medications is a crucial parameter for assessing their efficacy. Although the DTs of ODFs are not officially specified, they are typically reported to be within 30 s [23]. In vitro DTs of ODFs based on HA were reported to range from 78.87 to 92.24 s with varying catechin concentrations [7] and from 34 to 95 s with varying calcium concentrations [45], slightly longer than those of pullulan-based films (18 s) [39] or gelatin-based films (20.4 s) [6]. ODFs with a DT exceeding 30 s might be more suitable for alleviating dry mouth, as noted by Kweon and Han [9], or for treatments requiring prolonged efficacy within the oral cavity.

### 3.6. Vitamin C Content in a Sheet of Film

Table 3 shows the vitamin C content in the ODF measured by HPLC. The vitamin C content in one film (2 × 3 cm) showed a significant difference according to casting conditions, and CT2, CT4, CTD4, and CT8 contained 55.62, 59.98, 65.76, and 99.28 mg/film area (6 cm^2^), respectively. This is because the weight of the film increased depending on the casting thickness, despite having the same area (6 cm^2^), resulting in a higher amount of vitamin C contained within the film. In the case of CT8, it was found that consumption of one sheet (2 × 3 cm) satisfies the recommended intake amount for male and female adults (90 and 75 mg/day, respectively) presented by the US FDA, and CT2, CT4, and CTD4+4 also satisfy the recommended intake (45 mg/day) of the WHO with one sheet.

### 3.7. Vitamin C Release Aspect by Solvent

The sublingual mucosa to which ODF adheres is thin and also contains large veins. Thus, ODFs upon attachment to the sublingual mucosa directly enter the bloodstream. This facilitates rapid absorption of active ingredients, contributing to immediate bioavailability [46]. Figure 3A shows how vitamin C is released from ODFs of varying thickness in simulated saliva fluid (SSF), which is the primary delivery pathway. During the initial 5 min, CT2 and CTD4+4 exhibited the fastest release, with this rapid trend persisting until 60 min, reaching a final release rate of 98%. CT4 and CT8 showed decreases in release extent with 4–6% released at 5 min and 91% and 87% at 60 min, respectively. Despite being thicker than CT4, CTD4+4 seemed to induce rapid release of active ingredients due to factors such as space between bilayers and uneven surfaces, as described in Section 3.1 and Section 3.2. 

To compare the release characteristics on the pH of the release medium, the dissolution rates were measured in simulated gastric and intestinal fluid and water, as well as in buffer solution. In SGF (pH 1.2, (B)), which is the lowest pH environment, the variation in dissolution patterns among the samples was more significant, but in other solvents, the samples exhibited relatively similar dissolution characteristics, indicating a tendency for decreased release of vitamin C as thickness increased. The dissolution rate in all solvents increased gradually, and in all solvents except SGF, the dissolution rate was over 80% after 60 min; CT2 and CTD4+4 in SIF in particular showed a high elution of more than 97%. CT8, however, showed the lowest dissolution characteristics (under 90%) in all solvents until 60 min of the experiment. Through the above experiments, most of the vitamin C content in the ODF could be introduced into the body, and the thinner the film, the faster the elution of the components in the film. Between the same casting thickness, double-casted film (CTD4+4) showed a maximum of 10–18% higher dissolution than that in the single-casted one. Thus, it was confirmed that the elution pattern of the active ingredient in the ODFs can be controlled by manufacturing the film by varying the thickness and number of castings.

### 3.8. Applied Model Formula Analysis

In order to identify the in vitro dissolution mechanism of the vitamin C in the ODF, four types of release kinetics models, namely the zero-order release model, Higuchi model, Hixson–Crowell cube root law, and Korsmeyer–Peppas model were applied, respectively, and the dissolution rate constant of each model, the correlation coefficient, and dissolution index were obtained and shown in Table 4. If the release of vitamin C in the film is controlled by the expansion of the film material, it will be suitable for zero-order kinetics, if it is due to the permeation of the eluate, it will follow the Hixson–Crowell cube root law, and if the release is made by porous penetration of the eluent, the Higuchi model will be followed, so the release mechanism of vitamin C according to the thickness of the film can be identified through various applications of the model [47,48]. The R2 values of the Higuchi model decreased in the range of 0.916–0.993 as pH decreased, but showed no significant difference based on thickness. On the other hand, the Hixson–Crowell cube root law suggests that the release of vitamin C from the film occurs due to the dissolution of the film caused by the penetration of the release medium [47]. The R2 values of the Hixson–Crowell cube root law ranged from 0.946 to 0.998, which was the highest among all the models, indicating that the release of vitamin C from HA-based ODFs can be considered most suitable for the Hixson–Crowell cube root law.

In order to analyze the effect of the diffusion of vitamin C in the gel and the disappearance of the gel containing it on the dissolution, the dissolution rate (K) and the dissolution index (n) were obtained by applying the Korsmeyer–Peppas model. The dissolution pattern and the dissolution behavior of the film was investigated, and the most suitable model was found through comparison of the correlation coefficient with each model. The R2 value of zero-order kinetics was 0.826–0.994, most of which were above 0.9, which means that the vitamin C release of the HA film is released at a constant rate regardless of the peristalsis of the gastrointestinal tract [47]. In addition, as the film thickness increased, the R2 value increased, showing a tendency to approach zero-order kinetics, and decreased as the pH of the elution solvent decreased. In the Korsmeyer–Peppas model, K represents the release rate and n represents the release exponent [22]. The Korsmeyer–Peppas model also exhibited high R2 values ranging from 0.920 to 0.999, and various n values ranging from 0.46 to 1.03. The n values generally decreased to some extent with decreasing pH and showed an increasing trend with thickness. Among these, when the n value is less than 0.45, it indicates Fickian diffusion release, meaning that diffusion is predominant in the release mechanism. 

When the n value ranges from 0.45 to 0.89, it represents non-Fickian release (anomalous release), indicating that both the gel diffusion of the active ingredient and gel erosion occur appropriately during the release. When the value is greater than 0.89, it signifies super case II transport, indicating that the erosion of the gel dominates in the release mechanism [49]. Therefore, the release of vitamin C from HA-based ODFs was found to exhibit slightly different release mechanisms depending on the thickness. In the case of CT2, CT4, and CTD4+4, the dominant release mechanisms of vitamin C were the diffusion of vitamin C within the HA gel matrix and the loss of HA containing vitamin C. On the other hand, for CT8, it was observed that the release mechanism of vitamin C was somewhat dominated by HA erosion-induced release of vitamin C, particularly in a low-pH solvent, namely simulated gastric fluid (SGF).

### 3.9. Sensory Evaluation

The sensory evaluation results of HA-based ODFs prepared with different casting thicknesses are presented in Table 5. There were no significant differences observed in the appearance, which ranged from 3.80 to 4.68 among all the films. In terms of sourness, CT4 and CTD4+4 showed significant differences with values of 5.24 and 4.08, respectively. This can be attributed to the film dissolution rate and vitamin C release rate influenced by the film thickness and casting number. For 8 mm ODFs, there were no significant differences in sourness and overall preference. CTD4+4 exhibited relatively rapid vitamin C release within the oral cavity due to its internal layered structure and microspaces. In the case of CT8, although it contains a significantly larger amount of vitamin C than CTD4+4, a continuous perception of sourness could occur during slow dissolution. 

In terms of mouth-feel properties, CT2 and CT8 received the highest and lowest scores of 4.24 and 2.82, respectively, showing significant differences based on the thickness (*p* < 0.05). This might be because the increase in film thickness resulted in an increased amount of hyaluronic acid per unit area, which led to the dissolution of hyaluronic acid in saliva, causing a sensation of a foreign body in the oral cavity. The overall preference score was higher for CT2 and CT4, with a score of 5.32 and 5.60, than CTD4+4 and CT8 (4.44 and 4.64, respectively) (*p* < 0.05). Based on these results, it can be concluded that the preference for HA-based ODFs containing vitamin C is influenced by thickness and vitamin C release rate, and to achieve a higher preference for the film, it is desirable to set the casting thickness to 4 mm.

## 4. Conclusions 

In this study, a hyaluronic acid-based orally disintegrating film (ODF) with varying casting thickness was examined as a convenient formulation for delivering vitamin C, and the physicochemical and dissolution characteristics of each film were compared. The casting thickness directly influenced the film properties; with increasing casting thickness, the film thickness, weight, and disintegration time increased, while the surface pH and disintegration rate decreased. Tensile strength also increased with film thickness, and the film with a laminated structure produced by double casting (CTD4+4) exhibited improved flexibility and elongation. The vitamin C content per film sheet increased with thickness. In the case of CT8, consuming one sheet provides approximately twice the World Health Organization’s (WHO’s) recommended intake of vitamin C. However, a prolonged dissolution time may result in an increased oral retention time and a perceived foreign body sensation.

The analyzed dissolution behavior was found to be most suitable for the Hixson–Crowell cube root law, and it was confirmed that the release of vitamin C in the film was due to porous penetration of the dissolution media. The release exponent of the Korsmeyer–Peppas model varied depending on the thickness of the film and the pH of the dissolution medium. As the film thickness increased or the pH of the dissolution medium decreased, the release exponent increased, indicating that the release of vitamin C is predominantly affected by the loss of hyaluronic acid in the dissolution medium. This study suggests that by adjusting the thickness of the film or number of castings, it is possible to control the release mechanism of the active components contained in the ODFs. According to the results of a sensory evaluation, the film cast with a thickness of 4 mm (CT4) received the most favorable ratings in terms of sensory attributes, while the laminated film was rated least favorably. Based on the findings of this study, it appears desirable to establish a casting thickness of 4 mm as the optimal manufacturing condition for the HA-based ODFs developed for delivering vitamin C.

## Figures and Tables

**Figure 1 foods-13-01418-f001:**
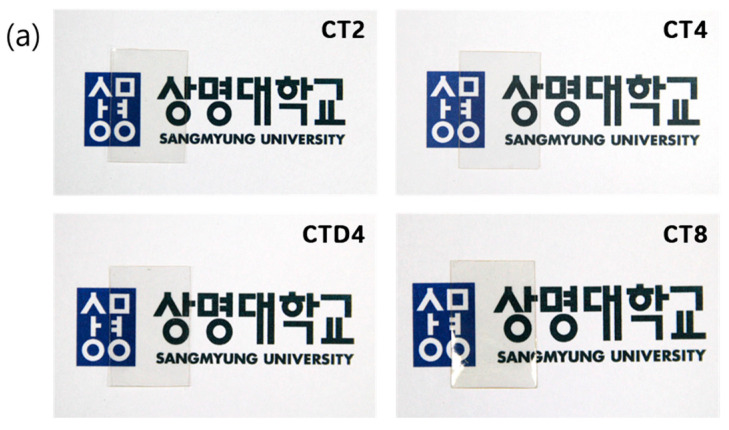
Appearance (**a**) and microstructure (surface and cross-section) (**b**) of each ODF. CT2: casting thickness 2 mm; CT4: casting thickness 4 mm; CTD4+4: casting thickness 4 + 4 mm (double casting); and CT8: casting thickness 8 mm.

**Figure 2 foods-13-01418-f002:**
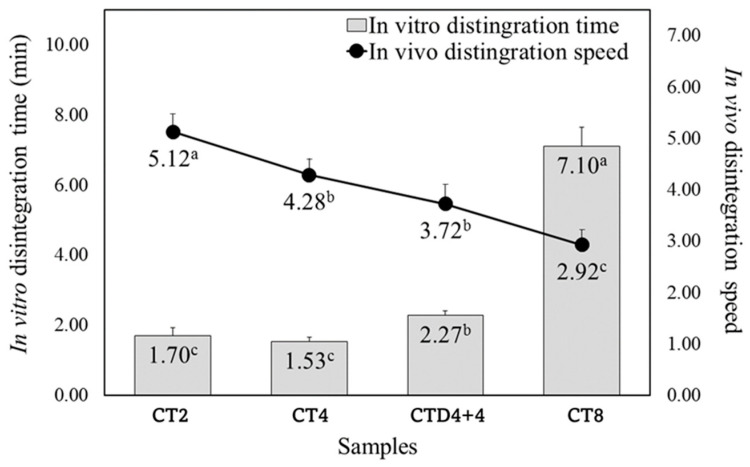
In vitro disintegration time and in vivo disintegration speed of HA-based ODFs with different casting thicknesses. Different letters represent significant differences (*p* < 0.05). All values represent mean ± S.D. CT2; casting thickness 2 mm; CT4; casting thickness 4 mm; CTD4+4; casting thickness 4 + 4 mm (double casting); and CT8; casting thickness 8 mm.

**Figure 3 foods-13-01418-f003:**
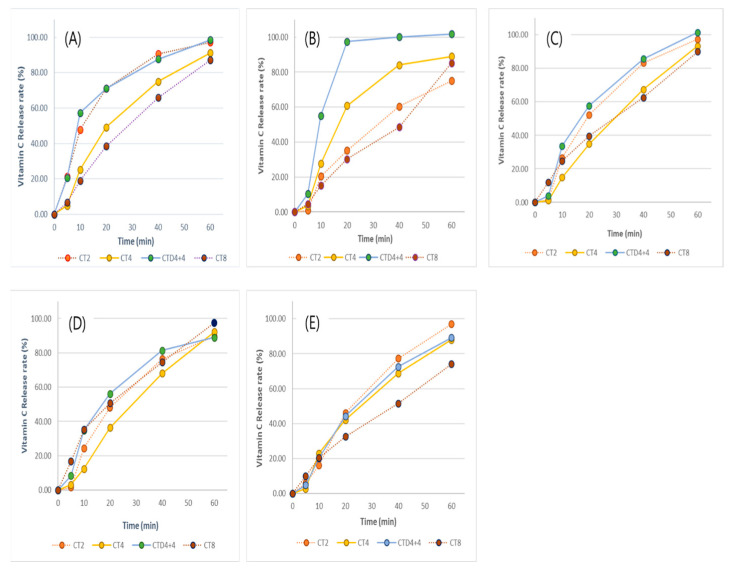
Dissolution profiles of vitamin C from HA-based ODFs in each dissolution media. (**A**) simulated saliva fluid (pH 6.8); (**B**) simulated gastric fluid (pH 1.2); (**C**) simulated intestinal fluid (pH 6.8); (**D**) acetate buffer (pH 4.5); and (**E**) distilled water. CT2: casting thickness 2 mm; CT4: casting thickness 4 mm; CTD4, casting thickness 4 + 4 mm (double casting); and CT8: casting thickness 8 mm. All values represent mean ± S.D.

**Table 1 foods-13-01418-t001:** Formula for dissolution medium (g, mL/L).

Chemicals	Dissolution Medium			
	Simulated Saliva Fluid	Simulated Gastric Fluid	Simulated Intestinal Fluid	Acetate Buffer
NaCl	8	2	-	-
KH_2_PO_4_	0.19	-	6.8	-
Na_2_HPO_4_	2.38	-	-	-
HCl	-	7	-	-
NaOH	-	-	0.616	-
C_2_H_3_NaO_2_	-	-	-	2.99
CH_3_COOH	-	-	-	1.66
Final pH	6.8	1.2	6.8	4.5

**Table 2 foods-13-01418-t002:** Effect of casting thickness on thickness, weight, surface pH, moisture content, and WVP.

Samples ^(1)^	Thickness(mm) ^(2)(3)^	Weight (mg/film)	Surface pH	Moisture Content (%) ^(2)(3)^	WVP (g·mm/m^2^·h·kPa)
CT2	0.09 ± 0.00 ^d^	115.40 ± 3.30 ^d^	4.05 ± 0.02 ^a^	16.62 ± 0.04 ^c^	5.67 ± 0.18 ^d^
CT4	0.14 ± 0.01 ^c^	124.73 ± 3.16 ^c^	3.60 ± 0.04 ^b^	18.94 ± 0.37 ^b^	7.07 ± 0.06 ^c^
CTD4+4	0.19 ± 0.01 ^b^	169.03 ± 1.59 ^b^	3.51 ± 0.01 ^c^	18.93 ± 0.66 ^b^	11.73 ± 0.14 ^a^
CT8	0.24 ± 0.01 ^a^	205.00 ± 5.06 ^a^	3.56 ± 0.02 ^c^	20.59 ± 0.35 ^a^	11.36 ± 0.13 ^b^

^(1)^ CT2: casting thickness 2 mm; CT4: casting thickness 4 mm; CTD4+4: casting thickness 4 + 4 mm (double casting); and CT8: casting thickness 8 mm. ^(2)^ All values represent mean ± S.D, *n* = 3. ^(3) a–d^ means within a column not followed by the same latter are significantly different at *p* < 0.05 of HA based-ODFs containing vitamin C.

**Table 3 foods-13-01418-t003:** Effect of casting thickness on transparency, mechanical properties, and vitamin C content of HA-based ODFs containing vitamin C.

Samples ^(1)^	Transparency	Mechanical Properties		Vitamin C Content (mg/Film)
		Tensile Strength (MPa)	Elongation at Break (%)	
CT2	20.05 ± 0.05 ^a(2)(3)^	6.28 ± 0.94 ^c^	25.34 ± 1.72 ^c^	55.62 ± 1.59 ^d^
CT4	14.31 ± 0.02 ^b^	6.97 ± 1.41 ^bc^	35.19 ± 5.88 ^b^	59.98 ± 2.05 ^c^
CTD4+4	10.42 ± 0.02 ^c^	16.61 ± 2.00 ^a^	54.08 ± 3.49 ^a^	65.76 ± 0.43 ^b^
CT8	8.04 ± 0.01 ^d^	7.99 ± 1.13 ^b^	38.81 ± 3.85 ^b^	99.28 ± 1.01 ^a^

^(1)^ CT2: casting thickness 2 mm; CT4: casting thickness 4 mm; CTD4+4: casting thickness 4 + 4 mm (double casting); and CT8: casting thickness 8 mm. ^(2)^ All values represent mean ± S.D, *n* = 5. ^(3) a–d^ within a column means significant difference at *p* < 0.05.

**Table 4 foods-13-01418-t004:** Release kinetics parameters with correlation coefficient for film containing vitamin C.

Samples ^(1)^	Dissolution Solution	Kinetic Models								
		Zero-Order Release		Higuchi Model		Hixson–Crowell Cube Root Law		Korsmeyer–Peppas Model		
		K	R^2^	K^1/2^	R^2^	K^1/3^	R^2^	K	n ^(2)^	R^2^
CT2	SSF	1.518	0.908	13.519	0.980	0.054	0.985	15.719	0.46	0.982
	SGF	1.796	0.899	15.659	0.950	0.105	0.975	11.357	0.57	0.950
	SIF	1.120	0.978	9.808	0.968	0.026	0.993	2.659	0.81	0.987
	PSB	1.579	0.967	13.051	0.968	0.043	0.995	4.351	0.75	0.981
	DW	1.554	0.985	12.652	0.966	0.042	0.998	2.908	0.85	0.990
CT4	SSF	1.567	0.972	12.954	0.976	0.044	0.998	4.764	0.73	0.987
	SGF	1.581	0.931	13.443	0.959	0.044	0.970	6.872	0.65	0.963
	SIF	1.633	0.995	12.938	0.955	0.046	0.989	1.661	0.99	0.995
	PSB	1.617	0.994	12.835	0.957	0.045	0.992	1.780	0.97	0.995
	DW	1.512	0.981	12.355	0.972	0.040	0.998	3.497	0.80	0.990
CTD4+4	SSF	1.479	0.890	13.309	0.970	0.055	0.983	17.852	0.43	0.974
	SGF	1.659	0.826	15.186	0.916	0.106	0.964	18.511	0.45	0.920
	SIF	1.737	0.959	14.537	0.973	0.091	0.962	6.351	0.69	0.981
	PSB	1.502	0.941	12.855	0.976	0.042	0.982	8.189	0.60	0.978
	DW	1.551	0.981	12.688	0.972	0.042	0.999	3.655	0.79	0.990
CT8	SSF	1.476	0.992	11.958	0.974	0.038	0.998	2.971	0.83	0.997
	SGF	1.377	0.994	10.829	0.947	0.035	0.972	1.199	1.03	0.994
	SIF	1.433	0.992	11.698	0.981	0.039	0.989	4.002	0.76	0.999
	PSB	1.526	0.976	12.823	0.993	0.052	0.987	7.718	0.62	0.998
	DW	1.116	0.919	9.632	0.961	0.025	0.946	7.715	0.55	0.961

^(1)^ CT2: casting thickness 2 mm; CT4: casting thickness 4 mm; CTD4+4: casting thickness 4 + 4 mm (double casting); and CT8: casting thickness 8 mm. ^(2)^ Release exponent. If n ≤ 0.45 means Fickian diffusion; 0.45 < n < 0.89 means anomalous transport; and n ≥ 0.89 means case-transport II.

**Table 5 foods-13-01418-t005:** Sensory attributes of hyaluronic acid-based vitamin C orally disintegrating films with various thickness.

Samples ^(1)^	Appearance ^(2)(3)^	Sourness ^(4)^	Sweetness	Mouth-Feel Properties	Overall Preference
CT2	4.60 ± 1.50 ^ns^	4.44 ± 0.92 ^ab^	4.44 ± 0.92 ^ns^	4.24 ± 1.09 ^a^	5.32 ± 1.46 ^ab^
CT4	4.68 ± 1.35 ^ns^	5.24 ± 1.05 ^a^	4.56 ± 0.92 ^ns^	3.68 ± 1.18 ^ab^	5.60 ± 1.04 ^a^
CTD4+4	3.80 ± 1.41 ^ns^	4.08 ± 1.38 ^b^	4.08 ± 1.08 ^ns^	3.40 ± 1.08 ^b^	4.44 ± 0.96 ^c^
CT8	4.28 ± 1.37 ^ns^	4.68 ± 1.25 ^ab^	4.44 ± 0.96 ^ns^	2.82 ± 1.28 ^c^	4.64 ± 1.47 ^bc^

^(1)^ CT2: casting thickness 2 mm; CT4: casting thickness 4 mm; CTD4+4: casting thickness 4 + 4 mm (double casting); and CT8: casting thickness 8 mm. ^(2) ns^ means within a column are not significantly different at *p* < 0.05. ^(3)^ All values represent mean ± S.D, *n* = 25. ^(4) abc^ means within a column not followed by the same latter are significantly different at *p* < 0.05.

## Data Availability

The original contributions presented in the study are included in the article, further inquiries can be directed to the corresponding author.

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
