# Peer review of "Influence of Casting Variables on Release Kinetics of Orally Disintegrating Film"

_foods, 2024, doi:10.3390/foods13091418_

Round 1
Reviewer 1 Report
Comments and Suggestions for Authors
Some revisions should be conducted.
1. The more key resutls data should be added in the abstract section.
2. The content of introduction setion can be extended by cited more related articles.
3. The introduction structure of indexes in the analysis methods, and resuts and discussion section should be revised and reproved. Such as 2.7. Mechanical properties and 2.6. Color and transparency should near to 2.4.
4. The results data discription should be writed in detail.
5. The effect mechanism and discussion should be improved by cited more articles.
Comments on the Quality of English Language
The English language should be revised and improved in the whole manuscript.
Author Response
Thank you for reading carefully and for your important and necessary comments. The responses to each comment are below and are marked in red in the text.
1.The more key results data should be added in the abstract section.
→ The differences in results according to the casting method was added, and unnecessary sentences was removed.
2. The content of introduction section can be extended by cited more related articles.
→ It seems that the necessary information is included in the introduction. Could you advise on any additional content that should be included in the introduction? The total number of references has already exceeded 45, and there is a concern that citing more references may lead to an excessive total number of references.
3. The introduction structure of indexes in the analysis methods, and results and discussion section should be revised and reproved. Such as 2.7. Mechanical properties and 2.6. Color and transparency should near to 2.4.
→ In the experimental methods section, the positions of mechanical properties and color and transparency were moved to near to 2.4. (thickness, weight, and surface pH), and the results section was also modified in the same order.
4. The results data discription should be writed in detail.
→ The interpretation of the results has been augmented, and the discussion has been enriched by incorporating additional references. Those were highlighted in red.
5. The effect mechanism and discussion should be improved by cited more articles.
→ It has been supplemented. Those were highlighted in red.
Reviewer 2 Report
Comments and Suggestions for Authors
This is an interesting study for formulating orally disintegrating films for delivery of vitamin c. The manuscript is prepared well with adequate background information and detailed methodology. The presentation of results also easy to follow. The following points need to be considered in the final draft
1. Page 2, lines 61-62: Provide the references of few studies
2. A more specific objective may add clarity to the paper.
3. Figure 1 is not clear. Provide higher resolution images.
4. Please try to provide a concise conclusion.
Author Response
Thank you for reading carefully and for your important and necessary comments. The responses to each comment are below.
Reviewer 2 comments
This is an interesting study for formulating orally disintegrating films for delivery of vitamin c. The manuscript is prepared well with adequate background information and detailed methodology. The presentation of results also easy to follow. The following points need to be considered in the final draft
- Page 2, lines 61-62: Provide the references of few studies
→ The introduction has been supplemented by adding references and marked in blue.
- A more specific objective may add clarity to the paper.
→ The purpose of this study was also supplemented in the introduction.
- Figure 1 is not clear. Provide higher resolution images.
→The picture presented is the original. As the size gets smaller, the resolution seems to decrease.
- Please try to provide a concise conclusion.
→The conclusion part has also been revised.
Reviewer 3 Report
Comments and Suggestions for Authors
Experimental
[1] Section 2.5 describes in vitro disintegration in water. Why not simulate oral saliva with one or more enzymes, particularly amylase?
[2] Section 2.7, slow-rate mechanical properties do not offer appropriate physical characteristic information of the films. Given the application and potential future development, tear strength properties would be more suitable.
[3] Section 2.9 describes in vitro dissolution using four different fluids. The premise of the manuscript is the development of a film containing vitamin C that will dissolve and release the vitamin in the oral cavity. In the introduction, it is suggested that ODFs dissolve in the mouth and the active ingredient is rapidly adsorbed to facilitate its delivery (lines 53-55). This suggests that the main delivery route for the vitamin C is oral rather than through the stomach. It is therefore unclear why dissolution is tested in gastric and intestinal fluids. Moreover, the simulated saliva fluid is only prepared using salts whereas it should include enzymes (see note regarding Section 2.5).
[4] The terms disintegration and dissolution are used to describe different mechanisms, but they are essentially describing the same process.
Results and Discussion
[5] Some photographs of the film would be appropriate for Section 3.1, particularly CT2.
[6] The bubbling could be reduced or eliminated with some further treatment.
[7] The SEM images are too small and poor quality. It is impossible to read the instrument information at the bottom of each image, especially the magnification or scale bar which is very important.
[8] Section 3.2, there is no discussion of the decreased total thickness of CTD4+4 in comparison to CTD8. It is unclear from the SEM cross section image whether the first layer was compressed by the second layer which could suggest it was not completely dry.
[9] Section 3.3, the relevance of this data is questionable (see earlier comments). In addition, the disintegration times seem to be excessively high for ODFs, what are other similar films reporting for this parameter?
[10] Section 3.4, WVP is not necessarily an important property for the application. Why is it included?
[11] Section 3.5, similarly color and transparency are not important for the application.
[12] Section 3.7, the unit of calculating the vitamin C content is ambiguous i.e. mg/film is not an appropriate unit. Either mg/unit area or mg/mg of film is a better measure.
[13] Sections 3.8-3.10 (release and release kinetics) need some further justification given the concerns of using appropriate solvents.
Conclusions
[14] Lines 532-534 are not specifically supported by the discussion. The statement also contradicts the final sentence of the abstract.
Comments on the Quality of English Language
Line 262 needs clarification of “an much amount”.
Author Response
Thank you for reading carefully and for your important and necessary comments. The responses to each comment are below and are marked in red in the text.
Experimental
[1] Section 2.5 describes in vitro disintegration in water. Why not simulate oral saliva with one or more enzymes, particularly amylase?
→ In-vitro test was conducted using water, given that water constitutes the primary component of saliva. Additionally, in-vivo test was also performed to assess the impact of various components such as enzymes and electrolytes within saliva on disintegration of ODF. Since in vivo testing was conducted, we did not perform the tests using simulated saliva separately.
[2] Section 2.7, slow-rate mechanical properties do not offer appropriate physical characteristic information of the films. Given the application and potential future development, tear strength properties would be more suitable.
→ Referring to literatures analyzing the characteristics of ODF, we have measured the tensile strength. However, as suggested by the reviewer, tear strength is also considered a crucial attribute in ODF. We appreciate the insightful suggestion and will include tear strength measurements for comparison in our future film research.
[3] Section 2.9 describes in vitro dissolution using four different fluids. The premise of the manuscript is the development of a film containing vitamin C that will dissolve and release the vitamin in the oral cavity. In the introduction, it is suggested that ODFs dissolve in the mouth and the active ingredient is rapidly adsorbed to facilitate its delivery (lines 53-55). This suggests that the main delivery route for the vitamin C is oral rather than through the stomach. It is therefore unclear why dissolution is tested in gastric and intestinal fluids. Moreover, the simulated saliva fluid is only prepared using salts whereas it should include enzymes (see note regarding Section 2.5).
→ As functional health foods, capsules, tablets, and film needs to be evaluated for their release characteristics in various organs (such as the oral cavity, stomach, and intestines) according to the Korean Pharmacopoeia or the United States Pharmacopeia. In this study, we also measured the release profiles at different pH levels in each organ for comparison. However, as accurately pointed out by the reviewer, because the primary route of delivery for vitamin C is oral, we deleted the unnecessary section (Vitamin C release aspect by thickness) and supplemented section 3.7.
→ In addition to the references cited in this study, many literatures including the example provided below, adjust the pH of simulated saliva by adding salts during manufacturing, without consideration for enzymes or other components. So, we also prepared simulated saliva by referencing existing literature.
â‘ Levallois et al. (1998). In vitro fluoride release from restorative materials in water versus artificial saliva medium (SAGF).
② Pietrzyńska & Voelkel (2017). Stability of simulated body fluids such as blood plasma, artificial urine and artificial saliva.
[4] The terms disintegration and dissolution are used to describe different mechanisms, but they are essentially describing the same process.
→ I agree with the reviewer's opinion. However, in this study, 'disintegration' was measured to assess the degree of film breakdown, whereas 'dissolution' was used to measure the extent to which fully dissolved functional ingredients (vitamin C) emerge, hence distinguishing and utilizing these two terms.
Results and Discussion
[5] Some photographs of the film would be appropriate for Section 3.1, particularly CT2.
→ The photographs of the prepared films were presented in Fig. 1(a).
[6] The bubbling could be reduced or eliminated with some further treatment.
→ We plan to add degassing treatment to completely remove bubbles, especially in thicker film.
[7] The SEM images are too small and poor quality. It is impossible to read the instrument information at the bottom of each image, especially the magnification or scale bar which is very important.
→ As the picture was reduced, the text became too small and unreadable, so the scale bar was made bolder and the magnification was added.
[8] Section 3.2, there is no discussion of the decreased total thickness of CTD4+4 in comparison to CTD8. It is unclear from the SEM cross section image whether the first layer was compressed by the second layer which could suggest it was not completely dry.
→ In section 3.1 and 3.2, each explanation has been supplemented.
[9] Section 3.3, the relevance of this data is questionable (see earlier comments). In addition, the disintegration times seem to be excessively high for ODFs, what are other similar films reporting for this parameter?
→ About the disintegration time, the content has been supplemented in section 3.5.
[10] Section 3.4, WVP is not necessarily an important property for the application. Why is it included?
→ An explanation has been added regarding the importance of WVP and the reason for measuring it.
[11] Section 3.5, similarly color and transparency are not important for the application.
→As suggested by the reviewer, the color results were deleted. In the case of transparency, however, even if the thickness was the same, there was a difference depending on the casting method (single vs. double), so it was left as is.
[12] Section 3.7, the unit of calculating the vitamin C content is ambiguous i.e. mg/film is not an appropriate unit. Either mg/unit area or mg/mg of film is a better measure.
→ The unit of calculating the vitamin C content was revised to film area (6 cm2).
[13] Sections 3.8-3.10 (release and release kinetics) need some further justification given the concerns of using appropriate solvents.
→ In connection with question [3], sections 3.7 have been revised and supplemented.
Conclusions
[14] Lines 532-534 are not specifically supported by the discussion. The statement also contradicts the final sentence of the abstract.
→ Inaccurate sentences were deleted, and the content was revised and supplemented.
Round 2
Reviewer 3 Report
Comments and Suggestions for Authors
Dear Authors,
Thank you for your thorough and considered responses to my comments and queries.